# A Spectral Energy Distance
# for Parallel Speech Synthesis

**Alexey A. Gritsenko**[*][†]    **Tim Salimans**[*]    **Rianne van den Berg**
**Jasper Snoek    Nal Kalchbrenner**
{agritsenko,salimans,riannevdberg,jsnoek,nalk}@google.com
Google Research

## Abstract

Speech synthesis is an important practical generative modeling problem that has seen great progress over the last few years, with likelihood-based autoregressive neural models now outperforming traditional concatenative systems. A downside of such autoregressive models is that they require executing tens of thousands of sequential operations per second of generated audio, making them ill-suited for deployment on specialized deep learning hardware. Here, we propose a new learning method that allows us to train highly parallel models of speech, without requiring access to an analytical likelihood function. Our approach is based on a generalized energy distance between the distributions of the generated and real audio. This *spectral energy distance* is a proper scoring rule with respect to the distribution over magnitude-spectrograms of the generated waveform audio and offers statistical consistency guarantees. The distance can be calculated from minibatches without bias, and does not involve adversarial learning, yielding a stable and consistent method for training implicit generative models. Empirically, we achieve state-of-the-art generation quality among implicit generative models, as judged by the recently-proposed cFDSD metric. When combining our method with adversarial techniques, we also improve upon the recently-proposed GAN-TTS model in terms of Mean Opinion Score as judged by trained human evaluators.

## 1   Introduction

Text-to-speech synthesis (TTS) has seen great advances in recent years, with neural network-based methods now significantly outperforming traditional concatenative and statistical parametric approaches [39, 35]. While autoregressive models such as WaveNet [35] or WaveRNN [15] constitute the current state of the art in speech synthesis, their sequential nature is often seen as a drawback. They generate only a single sample at a time, and since audio is typically sampled at a frequency of 18kHz to 44kHz this means that tens of thousands of sequential steps are necessary for generating a single second of audio. The sequential nature of these models makes them ill-suited for use with modern deep learning hardware such as GPUs and TPUs that are built around parallelism.

At the same time, parallel speech generation remains challenging. Existing likelihood-based models either rely on elaborate distillation approaches [27, 36], or require large models and long training times [29, 22]. Recent GAN-based methods provide a promising alternative to likelihood-based methods for TTS [3, 22]. Although they do not yet match the speech quality of autoregressive models, they are efficient to train and employ fully-convolutional architectures, allowing for efficient parallel generation. However, due to their reliance on adversarial training, they can be difficult to train.

---

[*]Equal contribution. [†] Work completed as a Google AI resident.

To address these limitations we propose a new training method based on the *generalized energy distance* [10, 32, 33], which enables the learning of implicit density models without the use of adversarial training or requiring a tractable likelihood. Our method minimizes a multi-resolution spectrogram loss similar to previous works [37, 38, 9, 6], but includes an additional *repulsive term* that encourages diverse samples and provides a statistical consistency guarantee. As a result, our models enjoy stable training and rapid convergence, achieving state-of-the-art speech quality among implicit density models.

In addition to demonstrating our proposed energy distance on the speech model of Bińkowski et al. [3], we further propose a new model for generating audio using an efficient overlap-add upsampling module. The new model is faster to run, while still producing high quality speech. Finally, we show that our proposed energy distance can be combined with GAN-based learning, further improving on either individual technique. An open source implementation of our generalized energy distance is available at `https://github.com/google-research/google-research/tree/master/ged_tts`.

## 2 Related work on speech synthesis

Our task of interest is synthesizing speech waveforms conditional on intermediate representations such as linguistic and pitch features, as usually provided by a separate model in a 2-step process. Here we briefly review the related literature on this problem.

**Autoregressive models.** van den Oord et al. [35] proposed WaveNet, an autoregressive model that produces high-fidelity speech by directly generating raw waveforms from the input features. WaveNet is trained by maximizing the likelihood of audio data conditional on the aforementioned linguistic and pitch features. While WaveNet's fully convolutional architecture enables efficient training on parallel hardware, at inference time the model relies on an autoregressive sampling procedure, generating the waveform one sample at a time. This necessitates tens of thousands of sequential steps for generating a single second of audio, making it ill-suited for real-time production deployment. These limitations were partially alleviated by Kalchbrenner et al. [15]. While still autoregressive, using a single-layer recurrent neural network, weight sparsification and custom kernels, their WaveRNN model achieves faster-than-realtime on-device synthesis.

**Probability density distillation.** Parallel WaveNet [36] and ClariNet [27] used a trained autoregressive model such as WaveNet as a teacher network *distilling* its knowledge into a non-autoregressive likelihood student model that is trained to minimize the Kullback-Liebler (KL) divergence between student and teacher. Both approaches rely on an Inverse-Autoregressive Flow (IAF; Kingma et al. [19]) as a student network. The IAF is structured in such a way that, given a set of latents, the corresponding observables can be generated efficiently in parallel.

While the methods of Ping et al. [27] and van den Oord et al. [36] differ in the choice of distributions used in their models, they both found that optimizing the KL-divergence alone was insufficient for obtaining high-quality generations, and required careful regularization and auxiliary losses for the student models to converge to a good solution.

**Flow-based models.** To avoid having to use a two-stage training pipeline required by the distillation approaches, FloWaveNet [16] and WaveGlow [29] propose directly training a convolutional flow model based on the architectures of RealNVP [7] and Glow [18] respectively. These models can be trained using maximum likelihood, and approach the speech quality of WaveNet or its distillations. However, due to the limited flexibility of their coupling layers (the building blocks used to construct invertible models) flow-based approaches tend to require large networks. Indeed, both WaveGlow and FloWaveNet have $\approx 100$ convolutional layers, and are slow to train [37].

Concurrently to our work Ping et al. [28] have made great progress towards reducing the size of flow-based TTS models. Their WaveFlow model achieves high quality speech synthesis using 128 sequential steps (more than two orders of magnitude fewer than a fully autoregressive model) while maintaining a small parameter footprint.

**Implicit generative models.** To date, Generative Adversarial Networks (GANs; Goodfellow et al. [11]) are mostly applied to image generation, where they are known to produce crisp, high-quality

images using fully convolutional architectures and relatively small models [5]. Their application to speech synthesis may provide an alternative to probability distillation that has the potential to match it in terms of speech quality. Three recent approaches, MelGAN [22], GAN-TTS [3] and Parallel WaveGAN [38], made significant progress in this direction. While still not matching WaveNet or ClariNet in speech quality, these works have proven the feasibility of solving TTS with implicit generative models. However, due to the reliance on adversarial learning, GANs can still be difficult to train. MelGAN and GAN-TTS rely on carefully chosen regularization and an ensemble of discriminators to achieve stable training, and MelGAN and Parallel WaveGAN use various auxiliary losses. This training difficulty also limits the available model choice, as only certain types of models (e.g. with batch normalization and spectral normalization) are known to be trainable.

## 3  Maximum Mean Discrepancy and Energy Distance

Before we present our proposed training method in Section 4, we briefly review previous work on learning implicit generative models, and we introduce the primitives on which our method is built.

Although GANs have recently become the dominant method for training implicit generative models without tractable likelihood, another popular approach to learning these types of models is a class of methods based on minimizing *Maximum Mean Discrepancy* (MMD), defined as

$$D_{\text{MMD}}(p|q) = \sup_{f \in \mathcal{H}} \left[ \mathbb{E}_{\mathbf{x} \sim p(\mathbf{x})}[f(\mathbf{x})] - \mathbb{E}_{\mathbf{y} \sim q(\mathbf{y})}[f(\mathbf{y})] \right], \tag{1}$$

where $f$ is a critic function which is constrained to a family of functions $\mathcal{H}$ [see 12], and $p(\mathbf{x})$ and $q(\mathbf{y})$ are the data and model distributions respectively. When $\mathcal{H}$ is given by a family of neural network discriminators, MMD becomes very similar to GANs, as explained by [2]. The main difference is that for MMD the maximization over $f \in \mathcal{H}$ is assumed to be analytically tractable, while GANs maximize over $f$ approximately by taking a few steps of stochastic gradient descent. The benefit of exact optimization is that MMD methods are provably stable and consistent, unlike GANs, although this comes at the cost of more restrictive critic families $\mathcal{H}$.

Gretton et al. [12] show that exact optimization is indeed possible if $\mathcal{H}$ is chosen to be a reproducing kernel Hilbert space (RKHS). In that case, there exists a kernel function $k \in \mathcal{H}$ such that every critic $f \in \mathcal{H}$ can be expressed through its inner product with that kernel:

$$f(\mathbf{x}) = \langle f, k(\cdot, \mathbf{x}) \rangle_{\mathcal{H}} = \sum_i \alpha_i k(\mathbf{x}, \mathbf{x}_i). \tag{2}$$

In other words, $f$ is constrained to be a weighted sum of basis functions $k(\mathbf{x}, \mathbf{x}_i)$, with weights $\alpha$. Exact optimization over $\alpha$ then gives the following expression for the squared MMD:

$$D_{\text{MMD}}^2(p|q) = \mathbb{E}[k(\mathbf{x}, \mathbf{x}') + k(\mathbf{y}, \mathbf{y}') - 2k(\mathbf{x}, \mathbf{y})], \tag{3}$$

where $\mathbf{x}, \mathbf{x}' \sim p(\mathbf{x})$ and $\mathbf{y}, \mathbf{y}' \sim q(\mathbf{y})$ are independent samples from $p$ and $q$.

Since (3) only depends on expectations over $q$ and $p$ it can be approximated without bias using Monte Carlo sampling. If our dataset contains $N$ samples from $p(\mathbf{x})$ and we draw $M$ samples from our model $q(\mathbf{y})$, this gives us the following stochastic loss function [12]:

$$L(q) = \frac{1}{N(N-1)} \sum_{n \neq n'} k(\mathbf{x}_n, \mathbf{x}_{n'}) + \frac{1}{M(M-1)} \sum_{m \neq m'} k(\mathbf{y}_m, \mathbf{y}_{m'}) - \frac{2}{MN} \sum_{n=1}^{N} \sum_{m=1}^{M} k(\mathbf{x}_n, \mathbf{y}_m). \tag{4}$$

Loss functions of this type were used by [8, 23] and [4] to train generative models without requiring a tractable likelihood function.

An alternative view on MMD methods is in terms of *distances*. As explained by Sejdinovic et al. [32], the kernel $k(\cdot, \cdot)$ of a RKHS $\mathcal{H}$ induces a distance metric $d(\cdot, \cdot)$ via

$$d(\mathbf{x}, \mathbf{y}) = \frac{1}{2}(k(\mathbf{x}, \mathbf{x}) + k(\mathbf{y}, \mathbf{y}) - 2k(\mathbf{x}, \mathbf{y})). \tag{5}$$

Assuming that $k(\mathbf{x}, \mathbf{x}) = k(\mathbf{y}, \mathbf{y}) = c$ with $c$ being a constant, equation (3) can equivalently be expressed in terms of this distance:

$$D_{\text{MMD}}^2(p|q) = D_{\text{GED}}^2(p|q) = \mathbb{E}[2d(\mathbf{x}, \mathbf{y}) - d(\mathbf{x}, \mathbf{x}') - d(\mathbf{y}, \mathbf{y}')], \tag{6}$$

which is known as the *generalized energy distance* [GED; see e.g. [32, 33, 30]].

In most practical applications of generative modeling, such as speech synthesis, we are interested in learning *conditional* distributions $q(\mathbf{x} \,|\, \mathbf{c})$ using examples $\mathbf{x}_i, \mathbf{c}_i$ from the data distribution $p$. In such cases we usually only have access to a single example $\mathbf{x}_i$ for each unique conditioning variable $\mathbf{c}_i$. This means that we cannot evaluate the term $\mathbb{E}[d(\mathbf{x}, \mathbf{x}')]$ in (6). However, this term only depends on the data distribution $p$ and not on our generative model $q$, so it can be dropped during training. The training loss then becomes

$$L_{\text{GED}}(q) = \mathbb{E}[2d(\mathbf{x}, \mathbf{y}) - d(\mathbf{y}, \mathbf{y}')], \qquad (7)$$

with $\mathbf{y}, \mathbf{y}' \sim q(\cdot \,|\, \mathbf{c})$ independent samples from our model, conditioned on the same features $\mathbf{c}$. This type of loss was studied by Gneiting and Raftery [10] under the name *energy score*. They find that (7) is a *proper scoring rule*, i.e. it can lead to a statistically consistent learning method, if the distance metric $d(\cdot, \cdot)$ is negative definite. This result is more general than the consistency results for MMD, and also allows for the use of distances that do not correspond to reproducing kernel Hilbert spaces. We make use of this result for deriving our proposed learning method, which we present in Section 4.

# 4 A generalized energy distance based on spectrograms

We require a method to learn generative models that can sample speech in a small number of parallel steps, without needing access to a tractable likelihood function. The method we propose here achieves this by computing a *generalized energy distance*, or *energy score*, between simulated and real data, and minimizing this loss with respect to the parameters of our generative model. Here, we assume that our dataset consist of $N$ examples of speech $\mathbf{x}_i$, labeled by textual or linguistic features $\mathbf{c}_i$. Our generative model is then a deep neural network that takes a set of Gaussian noise variables $\mathbf{z}_i$, and maps those to the audio domain as $\mathbf{y}_i = f_\theta(\mathbf{c}_i, \mathbf{z}_i)$, with $\theta$ the parameters of the neural network. This implicitly defines a distribution $q_\theta(\mathbf{y} \,|\, \mathbf{c})$ of audio $\mathbf{y}$ conditional on features $\mathbf{c}$.

Given a minibatch of $M$ examples $\{\mathbf{x}_i, \mathbf{c}_i\}_{i=1}^{M}$, we use our model to generate two independent samples $\mathbf{y}_i = f_\theta(\mathbf{c}_i, \mathbf{z}_i)$, $\mathbf{y}_i' = f_\theta(\mathbf{c}_i, \mathbf{z}_i')$ corresponding to each input feature $\mathbf{c}_i$, using two independently sampled sets of noise variables $\mathbf{z}_i, \mathbf{z}_i'$. We then calculate the resulting minibatch loss as

$$L_{\text{GED}}^*(q) = \sum_{i=1}^{M} 2d(\mathbf{x}_i, \mathbf{y}_i) - d(\mathbf{y}_i, \mathbf{y}_i'), \qquad (8)$$

where $d(\cdot, \cdot)$ is a distance metric between samples. The minibatch loss $L_{\text{GED}}^*(q)$ is an unbiased estimator of the energy score (7), and minimizing it will thus minimize the generalized energy distance between our model and the distribution of training data, as discussed in Section 3.

In practice the performance of the energy score strongly depends on the choice of metric $d(\cdot, \cdot)$. When generating high-dimensional data, it is usually impossible to model all aspects of the data with high fidelity, while still keeping the model $q_\theta(\mathbf{y} \,|\, c)$ small enough for practical use. We thus have to select a distance function that emphasizes those features of the generated audio that are most important to the human ear. This is similar to how GANs impose a powerful visual inductive bias when modeling images using convolutional neural network discriminators. Following the literature on *speech recognition* [20, 1], we thus define our distance function over *spectrograms* $\mathbf{s}^k(\mathbf{x}_i)$, where a spectrogram is defined as the magnitude component of the short-time Fourier transform (STFT) of an input waveform, $|\text{STFT}_k(\mathbf{x}_i)|$, where $k$ is the frame-length used in the STFT. Following Engel et al. [9] we combine multiple such frame-lengths $k$ into a single multi-scale spectrogram loss. Our distance function to be used in the generalized energy distance then becomes

$$d(\mathbf{x}_i, \mathbf{x}_j) = \sum_{k \in [2^6, \ldots, 2^{11}]} \sum_t || \, \mathbf{s}_t^k(\mathbf{x}_i) - \mathbf{s}_t^k(\mathbf{x}_j) ||_1 + \alpha_k || \log \mathbf{s}_t^k(\mathbf{x}_i) - \log \mathbf{s}_t^k(\mathbf{x}_j) ||_2, \qquad (9)$$

where we sum over a geometrically-spaced sequence of window-lengths between 64 and 2048, and where $\mathbf{s}_t^k(\mathbf{x}_i)$ denotes the $t$-th timeslice of the spectrogram of $\mathbf{x}_i$ with window-length $k$. The weights $\alpha_k$ of the L2 components of the distance are discussed in Appendix A. As we show there, the analysis of Gneiting and Raftery [10, Theorem 5.1] can be used to show that this choice makes (8) a strictly proper scoring rule for learning $q_\theta(\mathbf{x} \,|\, \mathbf{c})$ with respect to the ground-truth conditional distribution over spectrograms, meaning that $L_{\text{GED}}(q) > L_{\text{GED}}(p)$ for any $q(\mathbf{s}^k(\mathbf{x}) \,|\, \mathbf{c}) \neq p(\mathbf{s}^k(\mathbf{x}) \,|\, \mathbf{c})$. Minimizing this easily computable loss, we thus obtain a stable and statistically consistent learning method.

## 4.1 Why we need the repulsive term

Spectrogram-based losses are popular in the literature on audio generation. For example, the probability distillation methods ClariNet [27] and Parallel WaveNet [36] minimize the distance between spectrogram magnitudes of real and synthesized speech in addition to their main distillation loss; and Pandey and Wang [26] use a spectrogram-based loss for speech enhancement. Multi-resolution spectrogram losses like ours were used previously by Wang et al. [37] and Yamamoto et al. [38] for speech synthesis, and by Engel et al. [9] and Dhariwal et al. [6] for music generation. The main difference between these approaches and our generalized energy distance (Equation 9) is the presence of a *repulsive term* between generated data in our training loss, $-d(\mathbf{y}_i, \mathbf{y}_i')$, in addition to the attractive term between generated data and real data, $d(\mathbf{x}_i, \mathbf{y}_i)$.

The presence of the repulsive term is necessary for our loss to be a proper scoring rule for learning the conditional distribution of audio given linguistic features. Without this term, generated samples will collapse to a single point without trying to capture the full distribution. For many purposes like speech and music synthesis it might be argued that a single conditional sample is all that is needed, as long as it is a good sample. Unfortunately the standard loss *without* the repulsive term also fails at this goal, as shown in Figure 1. If the conditional distribution of training data is multi-modal, regression losses without repulsive term can produce samples that lie far away from any individual mode (Figure 1a). Even if the conditional distribution of training data is unimodal, such losses will tend to produce samples that are atypical of training data when applied in high dimensions (Figure 1b).

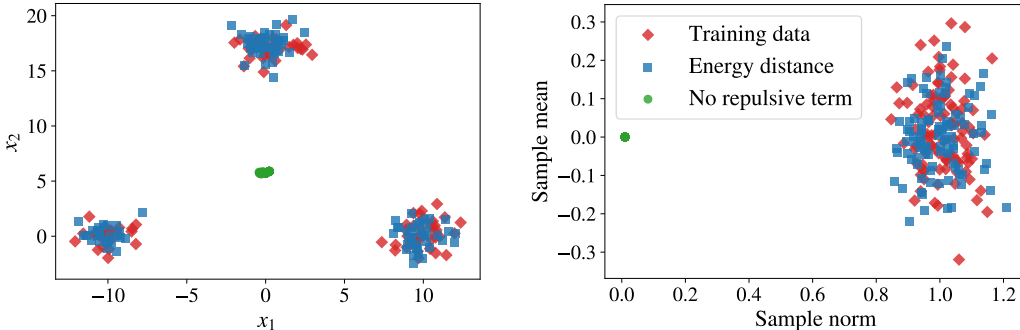

(**a**) Samples from a two-dimensional Gaussian mixture model with three components.

(**b**) Samples $\mathbf{x}$ from a single 100-dim Gaussian, with $||\mathbf{x}||_2$ on the x-axis and $\sum_i^n \mathbf{x}_i/n$ on the y-axis.

Figure 1: Samples from models trained by minimizing the energy distance (blue) or the more commonly used loss without repulsive term (green), and comparing to samples from the training data (red). Samples from the energy distance trained model are representative of the data, and all sampled points lie close to training examples. Samples from the model trained without repulsive term are not typical of training data. A notebook to reproduce these plots is included in our github repository.

In Section 7 we perform ablation experiments to further examine the role of the repulsive term for our specific application of speech synthesis. There, we show that this term is critical for achieving optimal performance in practice.

## 5 Model and training procedure

The models we train using the loss function we derived in Section 4 consist of deep neural networks that map noise variables to the audio domain, conditioned on linguistic features, that is $\mathbf{y}_i = f_\theta(\mathbf{c}_i, \mathbf{z}_i)$. This is similar to how conditional generator networks are usually parameterized in GANs, see e.g. BigGAN [5] for the analogous case where images $\mathbf{y}$ are generated from noise $\mathbf{z}$ and class labels $\mathbf{c}$. For the generator network $f_\theta$ we explore 2 different choices:

**Simplified GAN-TTS generator** To clearly demonstrate the effect that using the generalized energy distance has on model training, we attempt to control for other sources of variation by using a generator architecture nearly identical to that of GAN-TTS [3]. Specifically, we use a deep 1D convolutional residual network [13] consisting of 7 residual blocks (see Figure 3 of the

Appendix). Compared to the generator of GAN-TTS, we simplify the model by removing the Spectral Normalization [25] and output Batch Normalization [34], which we empirically find to be either unnecessary or hurting model performance.

**Inverse STFT architecture**   To experiment with the wider choice in generative models allowed by our training method, we additionally explore a model that makes use of the Short Time Fourier Transform (STFT) representation that is prevalent in audio processing, and which we also used to define the energy distance we use for training. This model takes in the features and noise variables, and produces an intermediate representation $\text{stft}_i = f_\theta(\mathbf{c}_i, \mathbf{z}_i)$ which we interpret as representing a STFT of the waveform $\mathbf{y}_i$ that is to be generated. Here, $f_\theta$ consists of a stack of standard ResNet blocks that is applied without any upsampling, and is therefore faster to run than our simplified GAN-TTS generator. We then linearly project $\text{stft}_i$ to the waveform space by applying an inverse STFT transformation, thereby upsampling $120\times$ in one step. The final output of this model is thus a raw waveform, similar to the (simplified) GAN-TTS model. Further details on this architecture are given in Appendix D.4.

**Training procedure**   All of our models are trained on Cloud TPUs v3 with hyperparameters as described in Table 5 of the Appendix. For each training example we generate two independent batches of audio samples from our model, conditioned on the same features, which are then used to compute our training loss. Our model parameters are updated using Adam [17]. Figure 2 explains this training procedure visually.

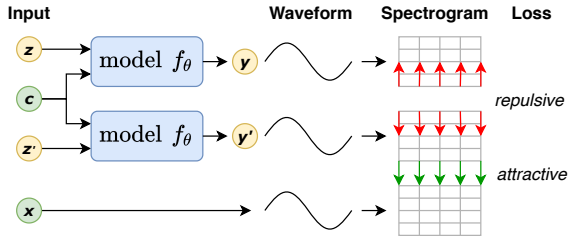

Figure 2: Visual depiction of our training process.

## 6   Data

Our TTS models are trained on a single-speaker North American English dataset, consisting of speech data spoken by a professional voice actor. The data consists of approximately sixty thousand utterances with durations ranging approximately from $0.5$ seconds to 1 minute, and their corresponding *aligned* linguistic features and pitch information. Linguistic features encode phonetic and duration information, while the pitch is given by the logarithmic fundamental frequency $\log F_0$. These features, a total of $567$, are used by our models as inputs and provide local conditioning information for generating the raw waveform. At training time, the features are derived from and aligned with the training audio. At test time, all features are predicted by a separate model; we thus never use ground-truth information extracted from human speech when evaluating our models.

To account for differences in utterance duration, during training we sample training examples with probability proportional to their length, and then uniformly sample 2 second windows and their corresponding features from these utterances. Examples shorter than 2 seconds are filtered out, leaving a total of $44.6$ hours of speech sampled at 24 kHz. The corresponding conditioning features are provided at 5ms intervals (200 Hz sampling frequency). The trained models are thus tasked with converting linguistic and pitch features into raw audio, while upsampling by a factor of 120. Our training setup is similar to that of Bińkowski et al. [3], except that we do not apply any transformations (e.g. a $\mu$-transform) to the waveforms.

## 7   Experiments

We evaluate our proposed approach to speech synthesis by training 4 different models on the data set described in the previous section:

1. The simplified GAN-TTS generator described in Section 5, but trained by minimizing our generalized energy distance.
2. This same model and loss, but leaving out the repulsive term $-d(\mathbf{y}_i, \mathbf{y}_i')$ as in previous works that use spectrogram-based losses.

3. Our inverse STFT model from Section 5 trained with the GED loss.

4. A hybrid architecture where we combine the GED loss with adversarial training using an ensemble of **un**conditional discriminators. Previous works like GAN-TTS and MelGAN use an ensemble of GAN discriminators with *conditional* discriminators taking in features $\mathbf{c}$, and *unconditional* discriminators that look only at the generated data $\mathbf{y}$. We hypothesize that our GED loss is a good substitute for the conditional part of the GAN loss, but that using an unconditional discriminator might yet offer additional benefit.

To evaluate our generated speech, we report the (conditional) Fréchet Deep Speech Distances (FDSD and cFDSD; Bińkowski et al. [3]) - metrics that judge the quality of synthesized audio samples based on their distance to a reference set. These distances are conceptually similar to the FID (Fréchet Inception Distance; Heusel et al. [14]) commonly used for evaluating GANs of natural images, but differ in that they (**i**) are computed on the activations of the Deep Speech 2 [1] speech recognition model in place of the activations of the Inception network [34] used in FID; and (**ii**) are computed for samples with conditioning features that match the reference set in the case of the conditional FDSD (cFDSD). We closely followed Bińkowski et al. [3] in our implementation of the FDSD metrics, but note several differences between the two implementations in Appendix E.1. We report these metrics on both the training data as well as a larger validation set. In addition, we also evaluate the quality of the synthesized audio using the Mean Opinion Score (MOS) computed from ratings assigned by trained human evaluators. These samples are generated on an *independent* set of 1000 out-of-distribution sentences for which no ground-truth data is available.

We compare our models against a careful re-implementation of GAN-TTS [3], built with help from the authors. Results are reported in Table 1 and include links to samples from each of our models.

Table 1: Mean Opinion Score (MOS) and (conditional) Fréchet Deep Speech Distances [3] (FDSD and cFDSD respectively) for prior work and the proposed approach. Models trained by minimizing our spectral generalized energy distance are indicated with GED. Our proposed generator using inverse STFT upsampling is marked iSTFT. For FDSD and cFDSD we report training scores for comparison to the numbers in Bińkowski et al. [3], as well as scores on the validation set. We truncate the sampling distribution of latents when generating from our models, as previously done in BigGAN [5]; we find this to give a slight boost in performance.

| MODEL | MOS | TRAIN FDSD | TRAIN cFDSD | VALID FDSD | VALID cFDSD | AUDIO SAMPLES |
|---|---|---|---|---|---|---|
| *Natural speech* | $4.41 \pm 0.06$ | 0.143 | | 0.156 | | |
| *Autoregressive models* | | | | | | |
| WAVENET [15] | $4.51^{\dagger} \pm 0.08$ | | | | | |
| WAVERNN [15] | $4.48^{\dagger} \pm 0.07$ | | | | | |
| *Parallel models* | | | | | | |
| MELGAN [22] | $3.72^{\dagger}$ | | | | | |
| PARALLEL WAVEGAN [38] | $4.06^{\dagger}$ | | | | | |
| GAN-TTS [3] | $4.21^{\dagger} \pm 0.05$ | 0.184 | 0.060 | | | |
| *Our models* | | | | | | |
| GAN-TTS *re-implementation* | $4.16 \pm 0.06$ | 0.163 | 0.053 | 0.193 | 0.077 | [Link] |
| GED *same generator* | $4.03 \pm 0.06$ | 0.151 | **0.020** | **0.164** | 0.038 | [Link] |
| GED *no repulsive term* | $3.00 \pm 0.07$ | 0.145 | 0.023 | 0.171 | 0.048 | [Link] |
| GED + iSTFT *generator* | $4.10 \pm 0.06$ | **0.138** | **0.020** | **0.164** | **0.037** | [Link] |
| GED + *unconditional* GAN | $\mathbf{4.25 \pm 0.06}$ | 0.147 | 0.030 | 0.169 | 0.040 | [Link] |

† Mean Opinion Scores reported by other works are included for reference, but may not be directly comparable due to differences in the data, in the composition of human evaluators, or in the evaluation instructions.

## 7.1 Discussion

**Spectral energy distance for TTS** We studied the effect that switching from adversarial training to training with the spectral energy distance has on the resulting models. To minimize the sources

of variation we used a generator architecture similar to that of the GAN-TTS (see Section 5 and Appendix 4). Table 1 shows that in terms of the cFDSD scores models trained with the GED loss improve by $\approx 2\times$ on the previously published adversarial model, GAN-TTS, suggesting that they are better at capturing the underlying conditional waveform distributions. We verified that this improvement is not due to overfitting by re-training the models on a smaller dataset (38.6 hours) and re-computing these metrics on a larger validation set (5.8 hours). We note, however, that the improved cFDSD scores did not transfer to higher subjective speech quality. In fact GED-only samples achieve lower MOS than the adversarial baseline, and empirically we found that FDSD metrics are most informative when comparing different versions of the same architecture and training setup, but not across different models or training procedures.

In two ablation studies (see Appendix B and C) of the components of our spectral energy loss we confirmed that the GED's repulsive term and the use of multiple scales in the spectral distance function are important for model performance. Moreover, a comparison between the results for GED and GED *no repulsive term* in Table 1 shows a significant decrease in MOS scores when the repulsive term is not present; and qualitatively, in the absence of the repulsive term, the generated speech sounds metallic. Since using spectrogram-based losses without the repulsive term is standard practice, we feel that comparison against this baseline is most informative in forecasting how useful the proposed techniques will be for the wider community.

**Combining GED and adversarial training**   The GED loss provides a strong signal for learning the conditional (local) waveform distribution given the linguistic features, but unlike GANs it does not explicitly emphasize accurately capturing the marginal distribution of speech. Empirically, we find that our GED-trained models can sometimes generate audio that, while perfectly audible and closely matching the original speech in timing and intonation, might still sound somewhat robotic. This suggests that these models might still benefit from the addition of an adversarial loss that specifically emphasizes matching the marginal distribution of speech. To test this, we trained the GAN-TTS architecture with its *conditional* discriminators replaced by a single GED loss. The resulting model (GED + *unconditional* GAN in Table 1) improves on the GED-only model as well as on GAN-TTS, achieving the best-in-class MOS of $4.25 \pm 0.06$.

**Choice of network architectures**   Encouraged by the stable training of our models with GED, we explored alternative architectures for speech synthesis, like our iSTFT generator (see Section 5 and Appendix D.4) that generates the coefficients of a Fourier basis and uses them within the inverse STFT transform to produce the final waveform. We find (GED + iSTFT *generator* in Table 1) that this architecture achieves the best training and validation (c)FDSD scores of the models we tried. In addition, it trains the fastest of our models, reaching optimal cFDSD in as little as 10 thousand parameter updates, with the per-update running time being about half that of the GAN-TTS generator. Unfortunately, this model did not significantly improve upon our results with the simplified GAN-TTS generator in terms of MOS. We tried using the iSTFT architecture in combination with adversarial learning but did not manage to get it to train in a stable way. This supports our belief that the spectral energy distance proposed in this work has the potential to enable the use of a much wider class of network architectures in generative modeling applications, and the design of novel architectures meeting the needs of specific applications (e.g. on-device efficiency).

**Train/test performance and overfitting**   Our models trained on the generalized energy distance are able to very quickly obtain good cFDSD scores after just 10 to 20 thousand parameter updates. When trained longer without any regularization, validation performance starts to deteriorate after that point. Unregularized, our models are able to produce samples on the training set that are very hard to distinguish from the data. We are actively working on developing new regularization techniques that more effectively translate this capacity into test set performance as measured by MOS.

## 8   Conclusion

We proposed a new generalized energy distance for training generative models of speech without requiring a closed form expression for the data likelihood. Our spectral energy distance is a proper scoring rule with respect to the distribution over spectrograms of the generated waveform audio. The distance can be calculated from minibatches without bias, and does not require adversarial learning, yielding a stable and consistent method for training implicit generative models. Empirical

results show that our proposed method is competitive with the state of the art in this model class, and improves on it when combined with adversarial learning.

Our proposed spectral energy distance is closely related to other recent work in audio synthesis [37, 38, 9, 6], in that it is based on calculating distances between spectrograms, but we believe it is the first to include a repulsive term between generated samples, and thus the first proper scoring rule of this type. We empirically verified that this is important for obtaining optimal generation quality in our case. Applying our scoring rule to the applications of these other works may offer similar benefits.

With the model and learning method we propose here, we take a step towards closing the performance gap between autoregressive and parallel generative models of speech. With further modeling effort and careful implementation, we hope that our method will be used to enable faster and higher quality generation of audio in live text-to-speech as well as other practical applications.

## Broader impact

The primary contributions of this paper introduce methodological innovations that improve the automated generation of speech audio from text. Positive aspects of automated text to speech could include improved accessibility for blind and elderly people or others who have poor eyesight. TTS is a cornerstone of assistive technology and e.g. is already used in the classroom to aid children with developmental disorders with reading comprehension. Although it is not within the scope of this work, automated TTS could be re-purposed to mimic a specific individual towards benevolent goals (e.g. to comfort someone with the voice of a loved one) or nefarious goals (e.g. to fake someone's voice without their permission).

## Funding disclosure

This work was funded by Google. None of the authors had financial relationships with other entities relevant to this work.

## Acknowledgments

We would like to thank Heiga Zen, Norman Casagrande and Sander Dieleman for their insightful comments, help with get acquainted with speech synthesis research and best practices, and for their aid with reproducing GAN-TTS results.

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
