[Supplementary Material]

# Appendix A  A proper scoring rule for speech synthesis

A loss function or *scoring rule* $L(q, \mathbf{x})$ measures how well a model distribution $q$ fits data $\mathbf{x}$ drawn from a distribution $p$. Such a scoring rule is called *proper* if its expectation is minimized when $q = p$. If the minimum is also unique, the scoring rule is called *strictly proper*. In the large data limit, a strictly proper scoring rule can uniquely identify the distribution $p$, which means that it can be used as the basis of a statistically consistent learning method.

In Section 4 we propose learning implicit generative models of speech by minimizing the *generalized energy score* [10] given by

$$L_{\text{GED}}(q, \mathbf{x}_i) = \mathbb{E}_{\mathbf{y}_i, \mathbf{y}'_i \sim q}\, 2d(\mathbf{x}_i, \mathbf{y}_i) - d(\mathbf{y}_i, \mathbf{y}'_i), \tag{10}$$

where $d$ is a distance function over training examples $\mathbf{x}_i$ and generated samples $\mathbf{y}_i, \mathbf{y}'_i$, both of which can be conditioned on a set of features $\mathbf{c}_i$.

In choosing $d()$, we follow the analysis of Gneiting and Raftery [10, Theorem 5.1, Example 5.7], who study the family of distance functions of the form $d(\mathbf{x}_i, \mathbf{x}_j) = ||\mathbf{x}_i - \mathbf{x}_j||_\alpha^\beta$ and prove that this choice makes (10) a proper scoring rule for learning $p(\mathbf{x})$ if $\alpha \in (0, 2]$ and $\beta \in (0, \alpha]$. This includes the special cases of L1 and L2 distance, the latter of which they show leads to a *strictly* proper scoring rule.

Given the restrictions set out by this analysis, and building on the domain-specific work of Engel et al. [9], we arrived at the following multi-scale spectrogram loss as our choice for $d$:

$$d(\mathbf{x}_i, \mathbf{x}_j) = \sum_{k \in [2^6, \dots, 2^{11}]} \sum_t ||\mathbf{s}_t^k(\mathbf{x}_i) - \mathbf{s}_t^k(\mathbf{x}_j)||_1 + \alpha_k ||\log \mathbf{s}_t^k(\mathbf{x}_i) - \log \mathbf{s}_t^k(\mathbf{x}_j)||_2, \tag{11}$$

where we sum over a geometrically-spaced sequence of STFT window-lengths between 64 and 2048, and where $\mathbf{s}_t^k(\mathbf{x}_i)$ denotes the $t$-th timeslice of the spectrogram of $\mathbf{x}_i$ with window-length $k$.

Rather than having a single scoring rule (10) combined with a multi-scale distance $d()$, we can equivalently rewrite our loss function as a sum over multiple scoring rules, each having a more simple distance function:

$$
\begin{aligned}
L_{\text{GED}}(q, \mathbf{x}_i) &= \sum_{k \in [2^6, \dots, 2^{11}]} \sum_t L_1^{k,t}(q, \mathbf{x}_i) + \alpha_k L_2^{k,t}(q, \mathbf{x}_i) \\
L_1^{k,t}(q, \mathbf{x}_i) &= \mathbb{E}_{\mathbf{y}_i, \mathbf{y}'_i \sim q}\, 2||\mathbf{s}_t^k(\mathbf{x}_i) - \mathbf{s}_t^k(\mathbf{y}_i)||_1 - ||\mathbf{s}_t^k(\mathbf{y}_i) - \mathbf{s}_t^k(\mathbf{y}'_i)||_1 \\
L_2^{k,t}(q, \mathbf{x}_i) &= \mathbb{E}_{\mathbf{y}_i, \mathbf{y}'_i \sim q}\, 2||\log \mathbf{s}_t^k(\mathbf{x}_i) - \log \mathbf{s}_t^k(\mathbf{y}_i)||_2 - ||\log \mathbf{s}_t^k(\mathbf{y}_i) - \log \mathbf{s}_t^k(\mathbf{y}'_i)||_2.
\end{aligned}
\tag{12}
$$

Here, each of the individual $L_1^{k,t}(q, \mathbf{x}_i)$ and $L_2^{k,t}(q, \mathbf{x}_i)$ terms is a proper scoring rule since it uses a L1 or L2 distance with respect to (the $\log$ of) the spectrogram slice $\mathbf{s}_t^k(\mathbf{x}_i)$. Furthermore, the sum of multiple proper scoring rules is itself a proper scoring rule, and it is strictly proper as long as at least one of the elements in the sum is strictly proper. This means that our combined loss $L_{\text{GED}}(q, \mathbf{x}_i)$ is indeed a strictly proper scoring rule with respect to $p(\mathbf{s}_t^k)$. It follows that it is also a proper scoring rule for $p(\mathbf{x} \,|\, \mathbf{c})$, but not necessarily a strictly proper one, since $\mathbf{x}$ may have long-range dependencies that cannot be identified from single spectrogram slices $\mathbf{s}_t^k$. We also experimented with adding such longer range terms to our training loss but found no additional empirical benefit.

We experimented with various weights $\alpha_k$ for the L2 term in (11), and found $\alpha_k = \sqrt{k/2}$ to work well. This choice approximately equalizes the influence of all the different L1 and L2 terms on the gradient with respect to our generator parameters $\theta$. Dropping the L2 terms by setting $\alpha_k = 0$ only gave us slightly worse results, and could be used as a simpler alternative.

For the calculation of the spectrograms $\mathbf{s}^k(\mathbf{x}_i)$ we obtained slightly better sounding results when mapping raw STFT outputs to the *mel*-frequency-scale, but with slightly worse results in terms of cFDSD. All reported results are with mel-scale spectrograms.

# Appendix B  Ablation study on the spectral energy distance

We carried out an ablation study, in which we systematically varied aspects of the spectral energy distance proposed in Section 4 while using the architecture described in Section 5. The results of

these ablations are presented in Table 2. We note that at a high level we observe that any deviation from the proposed spectral energy distance leads to higher (worse) values of the validation (c)FDSD metrics, and discuss specific ablation experiments below.

Compared to the Baseline GED model, the same model *without* the repulsive term ("Generalised energy distance: disabled" in Table 2) not only gets worse FDSD scores, but also significantly reduces quality of the synthesized speech (see Section 7), suggesting that this part of the loss is crucial for the models ability to accurately capture the underlying conditional waveform distributions.

We compute spectrograms using an overcomplete basis of sinusoids. An exploration of the effect of this oversampling ("DCT / DST overcompleteness" in Table 2) shows that the FDSD metric values stops improving beyond the use of an $8\times$ overcomplete basis. Another benefit of an overcomplete basis that is not captured by Table 2 is faster convergence of models with a more overcomplete basis; but this improvement too tapered off once the basis was at least $8\times$ overcomplete.

Finally, we explored the importance of using a multiple spectrogram scales in the GED loss ("Window sizes" in Table 2) by training models that each used only a single window size $k$ for its spectrograms. Our results show that individually all of the constituent window sizes yield worse results than when they are combined in a single loss, suggesting that use of multiple spectrogram scales is an important aspect of the proposed spectral energy distance.

Table 2: Validation FDSD metric values for experiments comparing the proposed model and its variants. The ablation experiments only ran for $200 \times 10^3$ training steps and not until convergence.

| STUDY | VARIANT | VALID FDSD | VALID CFDSD |
|---|---|---|---|
| Baseline GED | | 0.163 | 0.040 |
| Generalized energy distance | disabled | 0.170 | 0.047 |
| DCT / DST overcompleteness | 1x | 0.165 | 0.042 |
| | 2x | 0.165 | 0.041 |
| | 4x | 0.168 | 0.041 |
| | 16x | 0.163 | 0.041 |
| Window sizes | 64 | 0.195 | 0.087 |
| | 128 | 0.168 | 0.046 |
| | 256 | 0.166 | 0.043 |
| | 512 | 0.174 | 0.048 |
| | 1024 | 0.182 | 0.064 |
| | 2048 | 0.202 | 0.093 |

## Appendix C  Ablation study combining GED and GANs

On the suggestion of the reviewers we performed an additional ablation study to more carefully examine the interaction of an adversarial loss with our proposed GED loss. Table 3 shows cFDSD and MOS results for all combinations of 1) using a repulsive term or not, 2) using a multi-scale or single-scale spectrogram loss, and 3) using an unconditional GAN loss or not. These experiments ran for the full $10^6$ training steps, and include MOS scores as well as cFDSD scores, making them complimentary to the ablation study shown in Table 2.

The results in Table 3 confirm that including the repulsive term in the spectral energy distance always improves over the naive spectrogram loss in terms of MOS. Furthermore, we find that adding the adversarial loss is generally helpful, and that the multi-scale loss outperforms the single-scale loss.

Finally, we also ran an experiment combining our GED loss with GAN-TTS, with the conditional discriminators of GAN-TTS included. This experiment can be compared against the results in the main paper that only include *unconditional* discriminators when combining GED and GAN. As Table 4 shows, the combination of full GAN-TTS and GED performs about equally well as our proposed combination of GED and unconditional GAN. Both outperform the baseline of GAN-TTS without GED loss.

Table 3: Results for all combinations of (1) repulsive term (**r**) yes/no, (2) multi-scale (**m**) or single window size (256/512) or no spectrogram loss, (3) unconditional GAN loss (**G**) yes/no. Note that these ablations sampled the cFDSD validation set uniformly, where we used length-weighted sampling for the main paper and Table 4 below.

| model→ | **r+m+G** | **r+m** | **r+256+G** | **r+512+G** | **r+256** | **r+512** |
|---|---|---|---|---|---|---|
| cFDSD→ | 0.033 | 0.033 | 0.344 | 0.063 | 0.035 | 0.034 |
| MOS→ | $4.25 \pm 0.06$ | $4.06 \pm 0.06$ | $3.67 \pm 0.07$ | $3.96 \pm 0.059$ | $3.44 \pm 0.07$ | $2.89 \pm 0.09$ |
| model→ | **m+G** | **m** | 256+G | 512+G | 256 | 512 |
| cFDSD→ | 0.039 | 0.039 | 0.200 | 0.047 | 0.040 | 0.038 |
| MOS→ | $4.12 \pm 0.06$ | $3.00 \pm 0.07$ | $2.86 \pm 0.07$ | $3.82 \pm 0.06$ | $2.33 \pm 0.06$ | $2.48 \pm 0.06$ |

Table 4: Results for combining our proposed GED loss with full GAN-TTS, including the conditional discriminators, and comparing against GED + unconditional GAN, and GAN-TTS.

| model→ | GED + full GAN-TTS | GED + uncond. GAN | GAN-TTS only |
|---|---|---|---|
| cFDSD→ | 0.041 | 0.040 | 0.077 |
| MOS→ | $4.24 \pm 0.05$ | $4.25 \pm 0.06$ | $4.16 \pm 0.06$ |

## Appendix D    Training and architecture details

### D.1   Spectral distance

In practice, when computing the STFT spectrograms necessary for the spectral GED loss (9), we found that the training was more stable when spectrograms $s_i^k$ and $s_j^k$ were computed with Hann windowing, $50\%$ overlap and using an *overcomplete* Fourier basis. This is equivalent to transforming the windows of length $k$ using the Discrete Cosine and Discrete Sine (DCT and DST) with basis functions $\cos(\frac{2\pi}{k} \cdot \frac{i}{m})$ and $\sin(\frac{2\pi}{k} \cdot \frac{i}{m})$ to obtain the real and imaginary parts of the STFT, where $m$ is an integer oversampling multiplier and $i = 0, \dots, \frac{mk}{2} + 1$. For $m = 1$ this is equivalent to the standard Fourier transform, and we used $m = 8$ in our experiments. Importantly, we observed that using an $\times 8$ overcomplete basis did not significantly slow down training on modern deep learning hardware.

### D.2   Training details

Unless otherwise specified, all models were trained with the same hyper-parameters (see Table 5) on Cloud TPUs v3 with 128-way data parallelism and cross-replica Batch Normalization. Furthermore, unless otherwise specified, no additional regularization was used, i.e. the spectral energy distance was minimized directly. A single experiment took between 2 and 4 days to complete $10^6$ training steps.

GED + *unconditional* GAN used GAN-TTS hyper-parameters from Table 6, but with the generator learning rate set to $1 \times 10^{-4}$. The weight of the GED loss was set to 3.

GED + iSTFT *generator* used the Adamax [17] optimizer with $\beta_1 = 0.9$, $\beta_2 = 0.9999$, learning rate $10^{-3}$ with a linear warmup over 12000 steps, EMA decay rate of 0.99998 and early stopping to avoid overfitting.

### D.3   Simplified GAN-TTS generator

To convincingly demonstrate the usefulness of the generalized energy distance for learning implicit generative models of speech, we sought to compare it to GAN-TTS, a state-of-the-art adversarial TTS model. To this end in our core experiments we used an architecture that is nearly identical to that of the GAN-TTS generator, but is further simplified as described in Section 5 and as depicted in Figure 3.

Table 5: Default hyper-parameters.

| HYPER-PARAMETER | VALUE |
| --- | --- |
| Optimizer | Adam [17] |
| Adam $\beta_1$ | 0.9 |
| Adam $\beta_2$ | 0.999 |
| Adam $\epsilon$ | $10^{-8}$ |
| Learning rate | $3 \times 10^{-4}$ |
| Learning rate schedule | Linear warmup over 6000 steps |
| Initialization: shortcut convolutions | Zeros |
| Initialization: conditional Batch Normalization | Zeros |
| Initialization: rest | Orthogonal [31] |
| EMA decay rate | 0.9999 |
| Batch Normalization $\epsilon$ | $10^{-4}$ |
| Batch size | 1024 |
| Training steps | $10^6$ |

## D.4 Inverse STFT generator

Our inverse STFT generator takes in linguistic features **c** at a frequency of 1 feature vector per 120 timesteps (a *chunk*). A 1D convolution with kernel size 1 is used to project the features to a 2048 dimensional vector per chunk, which is then fed into a stack of 12 *bottleneck* ResNet blocks [13].

Each of the ResNet blocks consists of a kernel size 1 convolution to 512 channels, 2 convolutions of kernel size 5 at 512 channels, followed by projection to 2048 channels again. In-between the convolutions we use conditional batch normalization as also used in GAN-TTS and as described in Section E.

Finally we project down to 240 dimensions per chunk. Of these dimensions, one is used to exponentially scale the remaining 239 features. These remaining features are then interpreted as the non-redundant elements of an STFT with window size 240 and frame step 120, and are projected to the waveform space using a linear inverse STFT transformation. The model stack is visualized in Figure 5.

Figure 5: iSTFT model.

## Appendix E GAN-TTS baseline

We re-implemented the GAN-TTS model from Bińkowski et al. [3] for use as a baseline in our experiments. While attempting to reproduce the original implementation as closely as possible by following the description provided in [3], we observed that our implementation of the model (**i**) would not match the reported FDSD scores (reaching an cFDSD of $\approx 2.5$ instead of the reported 0.06); and (**ii**) would diverge during training. To alleviate these discrepancies, we found it necessary to deviate from the architecture and training procedure described in Bińkowski et al. [3] in several ways detailed below. Our modified implementation reaches cFDSD of 0.056 and trains stably.

**No $\mu$-transform.** We found that the use of a $\mu$-transform with 16-bit encoding ($\mu = 2^{16} - 1$) was the single largest factor responsible for low-quality samples in our initial implementation of GAN-TTS. With the $\mu$-transform enabled (i.e. generating and discriminating transformed audio), our GAN-TTS baseline converged very slowly and would only reach cFDSD of $\approx 2.5$ (see Figure 6). Disabling the $\mu$-transform was necessary for reaching competitive sample quality (0.056 cFDSD and 4.16 MOS). We also observed that the use of $\mu$-transform made training more unstable.

**Generator architecture.** We re-used most of the original generator architecture described in Bińkowski et al. [3], but empirically found that (**i**) adding a batch normalization followed by a non-linearity before the flower convolution; and (**ii**) switching from a kernel size 3 to a kernel size 1

(**a**) Generator architecture.

(**b**) Generator residual block (GBlock).

Figure 3: The proposed generative model (**a**) resembles the GAN-TTS generator and consists of 7 GBlocks (**b**) that use convolutional layers of increasing dilation rates, nearest-neighbour upsampling and conditional Batch Normalization. The number of channels is changed only in the block's first and shortcut convolutions; and the latter is only present if the block reduces the number of channels. The residual blocks follow the same upsampling pattern as GAN-TTS.

convolution; both led to more stable training with default settings. Addition of the former is inspired by the BigGAN architecture [5] that GAN-TTS is based on; and the latter relies on an interpretation of the first convolution as an embedding layer for the sparse conditioning linguistic features. These differences are reflected in the generator architecture in Figure 4a.

**Discriminator architecture.**    Empirically we found that it was necessary to introduce more changes to the discriminator architecture. Specifically, the following alterations were made (see also Figure 4b and Figure 4c):

- The mean-pooling along time and channel axes of the output of the final DBlock was replaced by a non-linearity, followed by sum-pooling along the time axis and a dense linear projection to obtain a scalar output. Like the addition of batch normalization and non-linearity before the generator output, this change is inspired by the BigGAN architecture.

- Instead of a single random slice of the input waveforms, each discriminator sampled *two* random slices $(\mathbf{x}_1, \mathbf{c}_1)$ and $(\mathbf{x}_2, \mathbf{c}_2)$, and produced independent outputs $d_1$ and $d_2$ for each of them. These outputs were later averaged to produce discriminators final output $d$.

(a) Generator.  (b) Conditional discriminator.  (c) Conditional discriminator residual block (DBlock).

Figure 4: Architectures used in our implementation of GAN-TTS. Generator (**a**) makes use of a smaller kernel size 1 convolution in the stem embedding linguistic features **c** and GBlocks identical to those in Figure 3. Discriminator (**b**) replaces mean-pooling with an additional non-linearity, sum-pooling and a final projection layer to obtain the scalars $d_{1,2}$ for each of the two random slices it samples. The random slice block takes aligned random crops (same for every example in the minibatch) $x_{1,2}$ and $c_{1,2}$ of the waveform **x** and conditioning features **c**, and the outputs for each of the two slices are averaged to obtain the final output $d = \frac{1}{2}(d_1 + d_2)$. Example architecture is shown for a conditional discriminator with window size 3600, but the same changes are applied to other window sizes and unconditional discriminators. The modified (conditional) DBlock (**c**) re-orders the first non-linearity and downsampling blocks.

- Inspired by the open source implementation of BigGAN[1], the structure of the first DBlock of each discriminator was altered to not include the first non-linearity. The architecture was surprisingly sensitive to this detail.
- Finally, the structure of the DBlocks was modified by (**i**) switching the order of the downsampling and non-linearity operations; and (**ii**) by reducing the dilation of the second convolution to 1 when the time dimension of the block is less or equal to 16.

**Hyper-parameters.** We recap all hyper-parameters used in our re-implementation of GAN-TTS in Table 6. As in the original publication, the GAN-TTS baseline was trained on a Cloud TPUs v3 with 128-way data parallelism and cross-replica Batch Normalization; training a single model took approximately 48 hours.

Training curves for our implementation of GAN-TTS, and how they compare to a similar (simplified) generator trained with the GED loss is shown in Figure 6.

Figure 6: GAN-TTS baseline (**our implementation**) with and without $\mu$-transform; and GED convergence speed and training stability.

Table 6: Hyper-parameters used in our implementation of the GAN-TTS baseline.

| HYPER-PARAMETER | VALUE |
|---|---|
| Optimizer | Adam [17] |
| Adam $\beta_1$ | 0 |
| Adam $\beta_2$ | 0.999 |
| Adam $\epsilon$ | $10^{-6}$ |
| Generator learning rate | $5 \times 10^{-5}$ |
| Discriminator learning rate | $10^{-4}$ |
| Learning rate schedule | Linear warmup over 6000 steps |
| Loss | Hinge [24] |
| Initialization | Orthogonal [31] |
| Generator EMA decay rate | 0.9999 |
| Batch Normalization $\epsilon$ | $10^{-4}$ |
| Batch Normalization momentum | 0.99 |
| Spectral Normalization $\epsilon$ | $10^{-4}$ |
| Batch size | 1024 |
| Training steps | $10^6$ |

## E.1 Fréchet Deep Speech Distances

At the time of writing no open source implementation of the Fréchet Deep Speech Distance (FDSD) metrics [3] was available. We thus resorted to re-implementating these metrics based on the information provided in the original publication. While striving to reproduce the original implementation as closely as possible, we deviated from it in at least two aspects, as discussed below.

Following the notation of Bińkowski et al. [3], let $\mathbf{a} \in \mathbb{R}^{48000}$ be a vector representing two seconds of (synthesized) waveform at 24 kHz; $\mathrm{DS}(\mathbf{a}) \in \mathbb{R}^{1600}$ be the sought representation that will be used for computing the (conditional) FDSD; and $f_{k\omega} : \mathbb{R}^{k\omega} \mapsto \mathbb{R}^{\lceil \frac{k}{2} \rceil \times 1600}$ be a function that takes (a part of) the waveform $\mathbf{a}$ and passes it through the pre-trained Deep Speech 2 (DS2) network [1, 21] to obtain the necessary activations. The representation $\mathrm{DS}(\mathbf{a})$ used for computing the Fréchet distance is then obtained by averaging the outputs of $f$ across time.

1. Equation (4) in Appendix B.1 of Bińkowski et al. [3] implies that the necessary activations were obtained *independently* for windows of the waveform $\mathbf{a}$ with window size $\omega = 480$ and step $\frac{\omega}{2} = 240$ (20ms and 10ms at 24kHz respectively), resulting 199 activation vectors

of size 1600 each, which were then averaged to obtain the representation DS($\mathbf{a}$). Doing so would not make any use of the DS2 bi-directional GRU layers, as their inputs would have time dimensionality of 1 - a single frame of the STFT with frame length $\omega$ and step $\frac{\omega}{2}$. So instead we used the entire audio fragment $\mathbf{a}$ (200 STFT frames) at once to obtain activations $f_{48000}(\mathbf{a}) \in \mathbb{R}^{100 \times 1600}$ that were averaged along the time axis to obtain DS($\mathbf{a}$).

2. Bińkowski et al. [3] proposed using activations from the node labeled `ForwardPass/ds2_encoder/Reshape_2` in the graph of a pre-trained DS2 network to obtain activations $f_{k\omega}$. This graph node belongs to the training pass of the model, and uses 6 layers with $0.5$ dropout probability (one after each of the 5 GRU layers, and then again after the last fully-connected layer of the encoder network), resulting very sparse activations. To make better use of the learned representations, we instead used the graph node labeled `ForwardPass_1/ds2_encoder/Reshape_2`, which implements the test time behaviour of the same network and produces dense activations.

The rest of the implementation followed Bińkowski et al. [3]. Namely, FDSD were estimated using $10000$ samples from the *training* data, matching the conditioning signals between the two sets in the case of conditional FDSD (cFDSD).

We tested our implementation by computing the FDSD for natural speech - the only quantity from Bińkowski et al. [3] that can be reproduced without access to a trained generator, and found that despite the implementation differences it agrees surprisingly well with the previously reported number (ours: $0.143$ vs. Bińkowski et al. [3]: $0.161$). We also considered implementations of the FDSD that did not deviate from the original description (i.e. using dropout and/or obtaining activations for each window independently), but found that they had worse agreement with the previously reported natural speech FDSD.

Without access to the original implementation it is impossible to tell whether there are other differences between the two FDSD implementations, or whether the described differences are actually there - the two implementations agree unexpectedly well on natural speech FDSD despite significant discrepancies in how they extract representations from the pre-trained model.

We hope that the difficulties we faced reproducing these results will prompt the research community to open-source evaluation metrics early on, even in cases when the models themselves cannot be made publicly available. We provide our implementation of FDSD in our github repository at `https://github.com/google-research/google-research/tree/master/ged_tts`.

## Appendix F  Mean Opinion Scores

Each evaluator, a native North American English speaker paid to perform the task was asked to rate the subjective naturalness of a sentence on a 1-5 (Bad-Excellent) Likert scale. Mean Opinion Scores (MOS) were obtained by summarizing as mean and standard deviation the $1000$ audio sample ratings produced by at least $80$ different human evaluators per test. The resulting scores are comparable between between the models trained in this work, but may not be directly comparable with previous work due to differences in composition of human evaluators and the evaluation instructions given to them.

## Appendix G  Linguistic features

As in [3, 15, 35, 36], synthesized speech was conditioned on local linguistic features and pitch information *predicted* from text using separate models; and ground truth linguistic features and pitch were used during training.

## Footnotes

[1]See https://github.com/ajbrock/BigGAN-PyTorch