[Reviews · NeurIPS 2020]

Review 1

Summary and Contributions: The author proposed a learning method that can generate speech fully in parallel, without explicit likelihood training. The proposed model is based on generalized energy distance between the distributions of the generated and real speech. This spectral energy distance can be calculated within mini-batches without bias and stabilize the training of implicit generative models. When combining the method with adversarial training, it generates high-quality speech in terms of Mean Opinion Score.

Strengths: 1. Stabilize the training process of implicit generative models using generalized energy distance; 2. Synthesize speech in parallel with high-fidelity in both GED and GED + GAN cases.

Weaknesses: The proposed generalized energy distance is similar with the normalization proposed in "Improved Techniques for Training GANs, Salimans et al., NIPS 2016". The "attractive" item in GED is similar with feature matching and the "repulsive" item in GED is similar with mini-batch discrimination, which is a normalization used for mode collapse. Further comparison is needed.

Correctness: The paper is technically solid.

Clarity: The paper is well written.

Relation to Prior Work: See comment above.

Reproducibility: Yes

Additional Feedback:


Review 2

Summary and Contributions: This work proposes the spectral energy distance for training parallel waveform models. It has an interesting repulsive term, which addresses the over-smoothing problem of spectrogram loss for high-fidelity speech synthesis. pros: - The repulsive term for spectrogram loss is well motivated and quite interesting. Overall, it is a complementary solution or even a good substitute for GAN loss, because it is much simpler to train. cons: - The ablation studies are not performed in a thorough way (see my comments).

Strengths: See my comments.

Weaknesses: See my comments.

Correctness: Yes.

Clarity: Yes.

Relation to Prior Work: Yes.

Reproducibility: No

Additional Feedback: Comments: - Section 2: Flow-based models are not necessarily large. The new SOTA WaveFlow is a small-footprint flow-based model for raw audio. The authors may reference WaveFlow and clarify the inaccurate claim in related work section. - After Eq.(1), one may mention the definition of p(x), q(y). - It is interesting that the repulsive term can provide such noticeable improvement of audio fidelity for spectrogram-based loss. - It would be more interesting to see an ablation study to investigate the individual contributions of repulsive term and multi-scale spectrogram loss. For example, what's the MOS score of combing repulsive term with single-scale spectrogram loss. I usually don't take such FDSD measures seriously, as it couldn't provide meaningful comparisons across different models in general, which is also observed by the authors. - As suggested by Parallel WaveGAN, combining multi-scale spectrogram loss with GAN loss could also provides good results. It would very nice to see an ablation study with MOS scores by varying three design choices: 1) w/ or w/o repulsive term, 2) single or multi-scale spectrogram loss, 3) w/ or w/o GAN loss. It will single out and emphasize the benefit of repulsive term under different circumstances. - The linguistic feature based TTS systems are almost infeasible to reproduce, as they involve huge amount of hand-engineered features. I wish the authors could provide as much information about these linguistic/pitch features as they can, for example, which model predicts these features at synthesis. In contrast, a neural vocoder experiment conditioned on mel spectrogram would be much easier to reproduce. ===================post rebuttal update============== I've read the rebuttal. I think the authors have spent substantial efforts to improve the paper from several aspects, including ablation studies, related work, reproducibility. Thus, I've changed the overall score from 6 to 7. Overall, I like this work. Note that, being more parallel / having fewer sequential steps is not necessarily an advantage. WaveFlow has more sequential steps than WaveGlow, but it runs faster on GPU at synthesis due to its small-footprint.


Review 3

Summary and Contributions: This paper derives a spectral distance for parallel TTS systems, and shows that when combined with GAN loss, it has a superior performance to GAN-based TTS systems.

Strengths: 1. The derivation of the proposed loss is clear and well-grounded. 2. The explanation of the repulsive term is clear and enlightening. 3. The generated audio samples are convincing.

Weaknesses: The major weakness of this paper is that the contribution is not among the most significant, nor is it well-supported by the experiments. Specifically: 1. Despite the well-grounded loss term, the proposed loss term is essentially Lp norm of the error in the frequency domain. The application of Lp norm on spectral error has been previously studied for many applications that aim to generate speech waveform, e.g. speech enhancement [1]. The repulsive term is a major novelty in such context, but this is not highlighted in the motivation, nor fully studied in the empirical evaluation. [1] Pandey, Ashutosh, and DeLiang Wang. "A new framework for CNN-based speech enhancement in the time domain." IEEE/ACM Transactions on Audio, Speech, and Language Processing 27.7 (2019): 1179-1188. 2. The performance advantage of the proposed metric over the baseline is not very pronounced. The main motivation is that the existing parallel synthesis baselines, including flow-based and GAN-based methods, are very hard to train, and the proposed loss is very easy to train. However, we do not see a competitive performance of using the proposed loss alone. The tradeoff between training complexity and performance is ubiquitous, and this paper fails to show that the proposed loss term achieves a good tradeoff. The only system that has a (marginally) better performance than the baselines is when it is combined with the GAN loss, but this defeats the purpose of having a simpler training scheme. 3. The authors claim that the proposed loss term is still useful even when it is combined with GAN, because it replaces the condition generation of GAN. However, this claim is unwarranted, because the authors did not compare the system with condition-GAN with that without. Thus it would be very helpful if the authors can also implement GED + conditional GAN. 4. There are other research attempts to find efficient AND high-quality synthesis architecture, e.g. WaveFlow [2], which is much easier to train and can still produce a high-quality synthesis. It would be nice if the authors can discuss the contribution in relation to such work. [2] Ping, Wei, et al. "Waveflow: A compact flow-based model for raw audio." arXiv preprint arXiv:1912.01219 (2019). 5. Considering the repulsive term is the major novelty, there should be a more thorough evaluation of the repulsive term. For example, it would be interesting to see the results of GED + GAN w/o repulsive term. If the repulsive term is truly irreplaceable, this result is expected to be much poorer than the best model. As a summary, the motivation, related work, and evaluation sections need to be improved in order to highlight the significance of the contribution of this paper.

Correctness: The derivation of the proposed loss term is correct. The empirical methodology is largely correct, although I remain skeptical of the FDSD metric. More specifically, since FDSD uses DeepSpeech2 features, and the input to the DeepSpeech2 is spectrogram, it is possible that any loss term that directly targets the spectrogram, such as the GED loss, would have an advantage under this metric. This can also explain why GED+iSTFT performs better than GED, because the former directly generates spectrograms, and thus has a greater degree of freedom to fit the distribution in the frequency domain. It would be helpful if the authors can comment on this.

Clarity: This paper has a clear explanation and is well-written.

Relation to Prior Work: It would be nice if the authors should discuss about the prior work that generates speech in the time domain but applies a spectral loss term. The authors should also include other TTS systems that attempt to simplify the training/generation procedure, e.g. WaveFlow. See 1 and 4 in the weakness review.

Reproducibility: Yes

Additional Feedback: I would like to thank the authors for the response. It addresses some of my concerns. Therefore, I have adjusted my score accordingly.

[Author Response · NeurIPS 2020]

We thank all reviewers for their input on our paper. All reviewers recognized the novelty and appeal of the repulsive term in our proposed loss, which is indeed the main technical contribution of this work. Since many previous works use spectrogram-based losses without such a term, and since adding the term greatly boosts performance, we also expect this contribution to have the greatest impact on future research.

The main improvement asked for by the reviewers is the addition of a set of ablation experiments to more clearly tease apart the contributions of the different loss terms and modeling choices. We're currently hard at work running and MOS testing all mentioned combinations of losses and models, and we report currently obtained results below. All results for all experiments proposed by the reviewers will be added to the final version of our paper.

**New ablations** (R3, R4) Partial results for requested new ablations. Missing MOS scores were not yet finished and will be added later.

| model→ | r+m+G | r+m | r+256+G | r+512+G | r+256 | r+512 | |
|---|---|---|---|---|---|---|---|
| cFDSD→ | 0.033 | 0.033 | still running | still running | 0.035 | 0.034 | |
| MOS→ | $4.25 \pm 0.06$ | $4.10 \pm 0.07$ | | | $3.44 \pm 0.07$ | $2.89 \pm 0.09$ | |
| model→ | m+G | m | 256+G | 512+G | 256 | 512 | G |
| cFDSD→ | 0.039 | 0.039 | 0.200 | 0.047 | 0.040 | 0.038 | 0.075 |
| MOS→ | $3.00 \pm 0.07$ | | | | | | $4.16 \pm 0.06$ |

Table 1: Showing results for all combinations of (1) repulsive term (**r**) yes/no, (2) multi-scale (**m**) or single window size (256/512) or no spectrogram loss, (3) GAN loss (**G**) yes/no. Note that these ablations sampled the cFDSD validation set uniformly, where we used length-weighted sampling for the submitted paper and the table below.

| model→ | GED + full GAN-TTS | GED + uncond. GAN | GAN-TTS only |
|---|---|---|---|
| cFDSD→ | still running | 0.040 | 0.077 |
| MOS→ | | $4.25 \pm 0.06$ | $4.16 \pm 0.06$ |

Table 2: Results for combining our proposed GED loss with full GAN-TTS, including the conditional discriminators (results pending), and comparing against GED + unconditional GAN, and GAN-TTS, as requested by R4.

**Small flow-based models do exist** (R3, R4) Our related work section now acknowledges that WaveFlow has made great progress in reducing the size of flow-based models while maintaining generation quality. A remaining advantage of our model is that it is still more parallel / has fewer sequential steps, as our models are fully convolutional while WaveFlow is a hybrid between autoregressive and parallel flow-based models.

**Limited performance gains against baseline** (R4) Our model with only the GED loss performs about the same as the SOTA GAN-TTS model from Binkowski et al. (MOS scores are not statistically different). However, our method is easier to train and converges faster. GED (which includes the repulsive term) also dramatically improves upon the baseline without repulsive term: Since using spectrogram-based losses without repulsive term is standard practice, we feel that comparison against this baseline is most informative in forecasting how useful the proposed techniques will be for the wider community.

**Repulsive term not highlighted or studied enough** (R4) Reviewer 4 notes that the repulsive term in our proposed loss is the main technical novelty in the paper, but that it should be highlighted and studied more. We now put even more emphasis on this contribution, and we add new ablation studies to better understand its impact (see above).

**GED+iSTFT benefits from directly generating spectrograms** (R4) Our iSTFT generator indeed has an inductive bias that might make it easier to do well on the cFDSD metric, however it also does very well on MOS. To avoid any possible confusion: the model still generates raw waveform audio, not just spectrograms.

**Relation to *Improved Techniques for Training GANs*** (R1) We now discuss the relationship between this paper and our proposed technique in the related work section. In short: the attractive term is indeed similar to the feature matching term in this paper if one would replace the discriminator activations of the feature matching loss with spectogram representations. Our repulsive term directly maximizes the distance in feature space between samples, whereas mini batch discrimination injects sample dissimilarity within a mini batch as side information into the discriminator which is trained with the usual discriminator loss of a GAN.

**Add definition of** $p(x), q(y)$ **after equation 1.** (R3) Done.

**New citation** *A new framework for CNN-based speech enhancement in the time domain***.** (R4) Added to related work section.

**More information on how to produce the linguistic features** (R3) Unfortunately the linguistic features that are used as conditioning input for our speech generation model, and which are generated from the source text by a separate model, are indeed not reproducible using publically available code. We now more fully describe these features in the appendix and, for comparison, we reference many papers that have used these features before.

[Meta-Review · NeurIPS 2020]

This paper proposes a strategy for parallel TTS based on spectral energy distance. It does not rely on explicit optimization of likelihood nor adversarial learning, which enjoys a more stable and consistent training. On top of that, the authors introduce a repulsive term which has shown to significantly improve the quality of the generated speech. When combined with adversarial training, the quality of speech can be further improved. Overall, this is an interesting work, technically solid and experimentally compelling. All reviewers are supportive for acceptance. The rebuttal is also pretty engaged with the comments and makes the work more convincing. Please finish up what is left in the rebuttal and revise the paper accordingly in the final version.